# Home Haemoglobin Monitoring for the Titration of Erythropoietin-Stimulating Agents in Chronic Kidney Disease: A Pragmatic Pilot Trial

**DOI:** 10.3390/diagnostics14020232

**Published:** 2024-01-22

**Authors:** Richard Bodington, Madeline Lok, Sunil Bhandari

**Affiliations:** 1Renal Research Department, Hull University Teaching Hospitals NHS Trust, Hull HU3 2JZ, UK; sunil.bhandari@nhs.net; 2Entia Ltd., 60 Gray’s Inn Road, London WC1X 8LU, UK

**Keywords:** eHealth, point-of-care-testing, anaemia, home-testing

## Abstract

Background. No trials of POCT device pathways have been published in the field of anaemia of CKD. We describe the results of a year of use of a novel POCT device with its associated eHealth pathway in the home monitoring of ESA therapy, with the aim of evaluating device performance and pathway feasibility. Methods: We used a POCT device designed for home self-testing, able to measure Hb from a drop of capillary blood (Luma, Entia, UK). Results were shared with HCPs via an associated mobile application. The pilot ran from August 2020 to March 2022 in a single UK renal centre. All adult non-dialysis-dependent-CKD patients on ESAs were eligible for inclusion. Participants were mailed the device and trained remotely. Participants were encouraged to self-test twice weekly for up to 1 year, with data collected on a pragmatic basis. Lab and Luma’s results were compared. Results: Seventeen participants returned comparable datasets (underwent ≥ 4 lab Hb measurements and self-tested for >5 months) with a mean testing frequency of 1.6 tests/wk. 1062 Luma and 113 lab Hb results were analysed. The coefficient of variation (CV) for raw results was 8.3% with a bias of −2.0% and SD 8.5. The percentage of Luma results differing by >10% lab results was 30.9%, dropping to 17.7% using an 8-point-moving-average. Participants stated they preferred Luma to traditional ESA monitoring and recommended the pathway to others. Conclusion: One year of home self-testing with Luma yielded comparable device utility to other POCT haematology analysers derived via HCP testing. Innovative pilot trials such as this form the basis for new empowering and home-based models of care required and desired by patients and HCPs.

## 1. Introduction

Chronic kidney disease (CKD) leads to reduced erythropoiesis and anaemia via the interrelated triad of erythropoietin deficiency, absolute and/or functional iron deficiency, and chronic inflammation [1,2]. Advanced CKD (CKD stages 4–5) affects 0.5% of the global population, and anaemia is seen in 50.3% of people with CKD 4 and 53.4% of those with CKD 5, representing a population of approximately 20 million people living with renal anaemia and advanced CKD worldwide [1,3].

Anaemia of CKD can be effectively treated with the supplementation of iron followed by the use of erythropoietin stimulating agents (ESA) or, more recently, the hypoxia-inducible factor (HIF) prolyl-hydroxylase inhibitors. The correction of severe anaemia (haemoglobin (Hb) < 90 g/L) has well-established benefits in terms of patient fatigue, physical functioning, the development of left ventricular hypertrophy, and mortality, in addition to the avoidance of blood transfusions with their associated risks [4]. However, the use of ESAs and HIF-prolyl hydroxylase inhibitors requires regular Hb monitoring to avoid the potential risks of stroke, cardiovascular events, venous thromboembolism, and hypertension associated with normalisation of Hb [4]. 

Point-of-care testing (POCT) can be performed by healthcare professionals (HCPs) or patients in a healthcare setting or in the patient’s home [5]. eHealth denotes the cost-effective and secure use of information and communication technologies, electronic patient records, telemedicine, and mobile health applications [6]. Patient self-testing in their own homes requires well-developed eHealth systems to support the use of POCT devices and to ensure patient safety [5]. The challenges of developing a POCT device suitable for home testing by patients in conjunction with eHealth solutions have resulted in only a small number of devices being approved for home use, the majority in the fields of diabetes and anticoagulation [5]. Although CKD is a common condition associated with high healthcare costs and burden to patients, virtually no home-testing pathways have been developed here despite the drive from a number of steering committees to develop more individualised and empowering patient pathways [5,7]. Empowering eHealth interventions, giving patients greater understanding and control over their condition, have been shown to improve outcomes in patients with CKD [8,9,10,11].

The home management of anaemia of CKD for those on ESAs or HIF- prolyl hydroxylase inhibitors represents one of the prime targets for home-testing pathways. Regular attendance at phlebotomy departments for the monitoring of Hb constitutes a substantial burden on these patients, who often suffer from several co-morbidities with their associated demands. Patients also have concerns regarding pain, damage to veins that may be later required for haemodialysis vascular access, and exposure to nosocomial infections, including COVID-19. Until now, no trials of devices and pathways in this space have been published. 

We describe the results of the use of a small handheld POCT device, Conformité Européene (CE), marked for home use, with its associated eHealth pathway, in the home monitoring of ESA therapy in patients with CKD at a kidney centre in the UK over a one year period, with the aim of evaluating device performance and pathway feasibility in this population.

## 2. Materials and Methods

In this pilot trial, we used a CE-marked POCT device designed for self-testing, able to measure Hb from a drop of capillary blood obtained via a finger prick (the Luma device, Entia, London, UK, CE certificate number 718734). Luma derives Hb concentration via haematocrit and mean corpuscular haemoglobin concentration in a 4–8 μL sample in 60 s by centrifugation followed by photometry. The device measures 78 × 83 × 52 mm and weighs 96 g (Figure 1). Capillary blood is drawn into a small plastic cuvette by capillary action loaded simply into the device, and a Hb result is obtained with a single button press. The device displays the Hb result on its integrated touch screen, and up to 32 results at once can be transferred to the associated mobile application via the photographing of a quick response (QR) code generated by the device. The Luma app allows the patient to view their cumulative Hb results and share the results with their HCPs, who are able to access results via email. The participant can navigate through the history of their results with the ability to annotate them with medication, symptoms, and side effect information (Figure 2). For patients who did not have a smartphone, the results could be read off the device to the HCP over phone calls. The Luma results were transferred into our kidney information management system (the Bradford, Hull, Leeds, and York (BHLY) system, VitalPulse, Chelmsford, UK), which allowed the results to be viewed separately, alongside, lab derived full blood count (FBC) results. 

The pilot trial ran from August 2020 to March 2022 in a single tertiary kidney centre in the UK (Hull University Teaching Hospitals NHS Trust, East Yorkshire, UK). All non-pregnant, non-dialysis-dependent kidney patients over 18 years of age on ESAs for the management of their anaemia of CKD were considered for inclusion in the trial. Candidates were suggested by the renal anaemia nurse specialist based on their individual suitability (dexterity, eyesight, and smartphone ownership) and gave informed written consent to participate. Participants received the Luma device via post and were trained remotely via video and phone calls by Entia staff; follow-up training was given as desired by the patient or required by Entia staff. Patients were encouraged to self-test twice weekly for up to 1 year, with data being collected on a pragmatic basis dependent on the patient’s treatment and preferences. If high levels of Luma and lab-derived Hb discrepancy were observed, an offer of re-training from Entia staff was given on an individualized basis. Given that the pilot trial used a CE-marked device as it was designed, formal research ethics committee approval was not required. The Trust Research Governance team was satisfied to grant a formal procurement contract funded by an Innovate UK grant. Luma results were collected by Entia via the Luma mobile app and shared with the NHS trust. The NHS trust supplied the lab-derived Hb results for the enrolled patients to Entia to allow analysis. Devices were returned to Entia at the close of the trial period. Traditional ESA monitoring using lab FBC continued during this period at a frequency in keeping with the local pre-existing protocol. The study did not protocolize a temporal relationship between the lab and Luma testing. Participants were invited for lab-based FBC tests at a frequency deemed necessary by the renal anaemia nurse with reference to local protocol.

Participant’s home-testing-derived Luma Hb results were compared to lab-derived Hb results (analysed on XN-10, Sysmex, Kobe, Japan) over the duration of the trial. Data on testing frequency, variability, and raw average differences between Luma Hb and lab Hb were collected alongside four- and eight-point moving average differences. Four and eight-point moving averages were used to reduce test-retest variation and aid HCP data interpretation. Coefficients of variation (CV) were calculated using outlier analysis using the interquartile range method and assuming a constant CV throughout the dataset. Analysis was performed on all data returned but was only felt to represent a meaningful comparison to the standard testing pathway for participants who tested for a sufficient period with several lab blood samples taken during this time. A comparable dataset included ≥4 lab Hb measurements and self-testing for >5 months. Feedback questionnaires were sent to all patients, and feedback was sought from HCPs involved in the use of the device. 

## 3. Results

From a total of 163 possible adult non-dialysis-dependent CKD patients requiring ESAs for the management of their anaemia of CKD, 48 patients (29.4%) were deemed appropriate for inclusion in and consented to the trial. The pilot trial generated 1515 Luma and 139 lab Hb results. 

Seventeen participants returned comparable datasets (underwent ≥ 4 lab Hb measurements and self-tested for >5 months); reasons for drop-out are detailed in the consort diagram of Figure 3. This population was used in the subsequent analysis and is henceforth referred to as the study population. The study population generated 1062 Luma and 113 lab Hb results (70% and 81% of the total data points generated in the study, respectively). Ninety-four percent of this group used Luma for >6 months. The mean lab Hb in the study population was 105.6 g/L. Table 1 summarises other aspects of Luma’s use and performance. The CV for raw results was 8.3% with a bias of −2.0% and a standard deviation (SD) of 8.5. 

In the study population, 31% of the raw Luma results differed by >10% from the lab results; this dropped to 17.7% with the application of an 8PMA (Table 1). This proportion was not meaningfully altered by the use of all data points from all 30 patients who used Luma (% raw Luma results differing by >10% lab results = 28.9%, % 8PMA Luma results differing by >10% lab results = 17.4%). 

Figure 4 gives a graphical representation of data for participants with low variability, high variability, and average variability throughout the testing period (Figure 4a, b, and c, respectively). Figure 4 also displays an example of a participant with high variability improving to low variability, suggesting an improved self-testing technique over time (Figure 4d).

Questionnaires were sent to all patients using the Luma device; nine (30%) responded. All participants who responded stated that they preferred Luma to the traditional method of monitoring their ESA and would recommend the pathway to others (Table 2). Feedback from HCP (*n* = 3) supporting participants using the device was predominantly positive. Positive comments focused on the preservation of veins, reduced patient travelling time, and potential exposure to COVID-19, with an appetite to continue using the device outside of a trial. Negative comments highlighted the fact that there remained the need for phlebotomy to check participants’ iron status and that some participants struggled to obtain a sufficiently large droplet of blood for analysis or to understand the inherent variation in haemoglobin results resulting from the device. 

## 4. Discussion

In this study, we present a trial of a home-based, self-testing, haematology POCT device. The trial recorded 1498 home Hb readings in a population where 70% of participants used the device for over 6 months. The use of single, ‘raw’, Luma Hb readings resulted in an unacceptably high proportion of data points outside 10% of the lab-derived Hb values; this was remedied using an 8-point moving average (8PMA) (% of Luma results >10% different from lab values = raw: 31% vs. 8PMA: 17.7%). This simple adaptation, using readings taken over an average of one month of testing, provided a level of accuracy appropriate for clinical decision making. The use of this metric does led to a lag in the identification of Hb trends, but the 8PMA rolling nature minimizes this. The mean 8PMA difference between Luma’s and lab-derived Hb values was 0.4 g/L (95% CI −0.4 to 1.2). In the lab setting, Luma demonstrates high correlation with a lab-based haematology analyser (LH 750, Beckman Coulter, Brea, CA, USA) (r = 0.99, coefficient of variation (CV) 7.1%) (unpublished data, Entia); in this pilot the Luma to the lab (XN-10, Sysmex) CV = 8.3%. This compares well with the standard POCT haematology analyser, the Hemocue Hb 301 system, recommended in the guidelines for Demographic and Health Surveys (DHS) for use by HCPs, and other haematology POCT devices, with CVs generally ranging from 2 to 8% [5,12]. The mean difference for venous samples for Hemocue 301 vs. lab-based measurement (Medonic M-series) was 6.9 g/L (95% CI 5.7 to 8.1) compared to Entia device vs. lab being 8.1 g/L (95% CI 7.3 to 8.8) albeit using different lab-based devices [12,13,14,15]. Similarly, the mean difference for venous samples between the Entia device and Hemocue was insignificant at −1.1 g/L (95% CI −2.3 to 0.0) [13]. The same study did show a significantly different mean difference between the Entia device and Hemocue utilising capillary blood samples (mean difference 3.2 g/L, 95% CI 2.2 to 4.2) [13]. This discrepancy highlights the well-described challenges associated with capillary blood sampling due to sampling technique, drop-to-drop variation, and patient factors, giving wider limits of agreement (LOA) and larger SD for devices using such samples compared to venous blood [13]. These limitations have led some evaluators of capillary POCT haematology devices to conclude that they cannot be reliably used to diagnose or monitor anaemia [13]. However, the potential limitations of a single capillary sample reading from such a device can be overcome by the use of repeated testing and multiple-point moving averages. Where this may be impractical with HCP-delivered POCT, home self-testing removes such barriers. However, the use of such a metric does lead to a lag between a true change in Hb and that result being detected. As such, some of the advantages of home testing are lost.

Comparison of variation between Luma and other devices, such as the HemoCue systems, is limited greatly by the fact that data from other devices are derived from HCP, and not home-testing. As far as the authors are aware, the only haematology POCT device with published data on use with patient self-testing is the HemoCue WBC DIFF. In a trial of 14 participants, WBC DIFF versus lab-derived white cell counts (WCC) found a mean difference (MD) WCC 0.36 × 10^9^/L, SD: 1.01, with r = 0.86, and 7.1% of measurement pairs outside LOA [16]. In a separate trial of the same device in n = 50, a high correlation between measurement pairs was demonstrated (HCP test versus patient test, R^2^ = 0.921, *p* < 0.001)] [17]. In both Hemocue studies, venous and capillary tests were taken within a few hours of each other. For this study, venous and capillary tests were separated by up to 3 days. This will have affected the CV reported between the capillary and venous tests. The reported mean 8PMA difference of 0.4 g/L (95% CI −0.4 to 1.2) in our study compares well, given the clear variability associated with patient self-testing. Similarly, the reported WBC DIFF home-testing CVs varied between 2.2 and 15.2% depending on the cell type analysed (overall WCC lowest CV and monocytes highest CV) is in keeping with the CV of 8.3% demonstrated with home-testing with Luma. 

In general, participants with high test-test variability, suggesting a non-optimal testing technique, remained poor throughout the trial, and those with low variability remained good. This observation hints at the limitations of repeated training sessions by video. There were select examples of participants who reduced their testing variability during the trial and those whose variability increased, as demonstrated in Figure 4. Training may have been more effective face-to-face for some participants, but these opportunities could not be offered during the COVID-19 pandemic in which this pilot took place. 

POCT devices alone are only one of many components required to implement a home POCT pathway. POCT pathways need to demonstrate accuracy, validity, and non-inferiority to traditional care [18]. Home POCT pathways introduce additional risks in terms of patient and staff training; both groups need training in interpreting results and the recognition of errors [19,20]. Data security also needs to be considered. Integration of POCT pathways into routine clinical care is costly and is liable to fail without dedicated support [21]. In this pilot trial, implementation of the Luma pathway was made possible by the cooperation of consultant nephrologists, a significant proportion of the full-time working hours of a research fellow, a research nurse, and two employees of Entia with some input from an NHS IT manager. This allowed the design of standard operating procedures, enrolment, training, distribution, integration of data into the renal information management system, and interpretation of Luma results. Although the requirement for staff hours would reduce after the set-up phase, a full-time member of staff would likely be required long-term to train and re-train patients if the pathway were offered to the entire renal anaemia cohort in our trust. The cost-effectiveness of such pathways, therefore, needs close attention; such analysis lies beyond the scope of the current paper. The York Health Economics Consortium (YHEC) has undertaken a preliminary economic analysis of the Luma device and has suggested that if the device replaced standard care for 75 patients under the care of Hull University Teaching Hospitals, 510 primary care appointments per year could be avoided and suggested cost neutrality of the Luma service (unpublished data, YHEC). A Cochrane meta-analysis of POCT technologies has shown a lack of evidence for cost-effectiveness [10]. However, studies have consistently demonstrated that home POCT pathways have potential positive outcomes with regard to quality of life, patient satisfaction, user acceptance, and patient empowerment [11]. The staff and patient feedback received in our study, albeit with limited numbers, confirm the positive views of both staff and patients with regards to Luma (patients rated their overall experience with Luma 4.9/5, 100% would recommend Luma to someone with similar health condition, 100% preferred Luma to traditional monitoring pathway, and staff comments positive and highlighted benefits for patients) (Table 2). Frequent home testing potentially provides additional benefits over traditional care by affording HCPs and patients better visibility of Hb trends, allowing earlier detection and expedited action in regard to adverse Hb trajectories. Such a case is demonstrated in Figure 4a; Luma demonstrated a falling Hb one month before a scheduled phlebotomy session. 

The current study has several limitations. Luma was deployed on a pilot basis; training was heavily supported by the device manufacturer in a limited population of patients for one year and was then completely withdrawn. As such, the current trial gives no evidence as to the cost-effectiveness of such a pathway or evidence of utility over a prolonged period. Only 29.4% of the anaemia of CKD cohort on ESAs were enrolled in the study; these patients were suggested in a non-random fashion by the renal anaemia specialist nurse on the basis of their interest or deemed suitability for the pilot. Such a method is clearly a source of bias in the results obtained; however, this decision was taken pragmatically on the basis that such a device, even in widespread use, would only ever be offered to those deemed capable of home testing. As such, a more randomised approach would potentially produce data less like that of the device’s widespread ‘real life’ use. Metrics such as CV and R^2^ compared to lab results, along with % results >10%, and mean difference from lab results varied widely between participants, presumably depending on capillary sampling technique. As such, the validity of the use of overall summaries of these metrics could be called into question. Additionally, capillary samples were never taken at the same time as venous samples; as such, other factors may have impacted the correlation between the Luma and lab results. 

This pilot trial forms a starting point for future work; full economic analysis is required, followed by detailed implementation work prior to a larger-scale roll-out of the pathway and careful service evaluation. 

## 5. Conclusions

NHS England makes clear the importance of giving patients more autonomy over their care in their ‘Long Term Plan’ and emphasizes the need to develop innovative models of care exploiting the information revolution [7,17,22]. Thirty patients with anaemia of CKD were trained to provide home haemoglobin measurements and provided 1498 home readings over 1 year at a mean frequency of 1.7 readings/wk. In the study population of 17 patients with datasets large enough for robust analysis, 1062 POCT Hb results were compared to 113 lab results; the bias between the POCT vs. lab was −2.0% with a CV of 8.3%. HCPs were given the option to view patient data in the form of raw, 4PMA, and 8PMA graphs for ease of interpretation. Both staff and patients responded positively towards the new home testing pathway, with all patients preferring the home testing pathway to traditional care. The Luma data suffered from the intrinsic variability associated with capillary blood sampling and displayed a widespread and large variation in precision between participants depending on the sampling technique. The pathway required the support of corporate partners to function, and full cost-effectiveness analysis is required. The use of Luma potentially allows a more rapid reaction of adverse haemoglobin trends and significantly reduces the need for phlebotomy with its associated drawbacks of travel time, exposure to nosocomial infections, and damage to veins, especially during the initiation of therapy when dose adjustments are more frequent. As such, the Luma pathway is preferred to traditional care in our small cohort. The Luma pathway does not remove the need to periodically check iron status via phlebotomy, and the 8PMA mandated by the variation in individual Hb results leads to a lag in the interpretation of Hb trends. Despite the challenges required to integrate and upscale such home-testing pathways, new models of care, empowering, home-based, and utilising the advances in information and communications technology, are clearly required and desired by both patients and healthcare professionals; pilots such as our own form a basis for this work. 

## Figures and Tables

**Figure 1 diagnostics-14-00232-f001:**
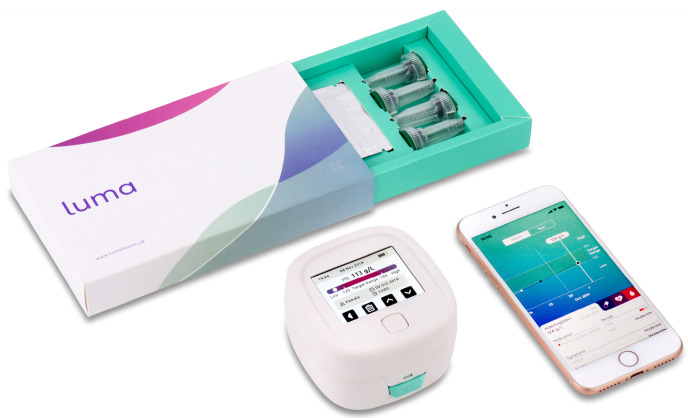
Entia’s ‘Luma’ home haemoglobin monitoring solution. Source: https://www.lumahealth.uk, accessed on 26 January 2023. Image used with written permission from Entia Ltd., London, UK.

**Figure 2 diagnostics-14-00232-f002:**
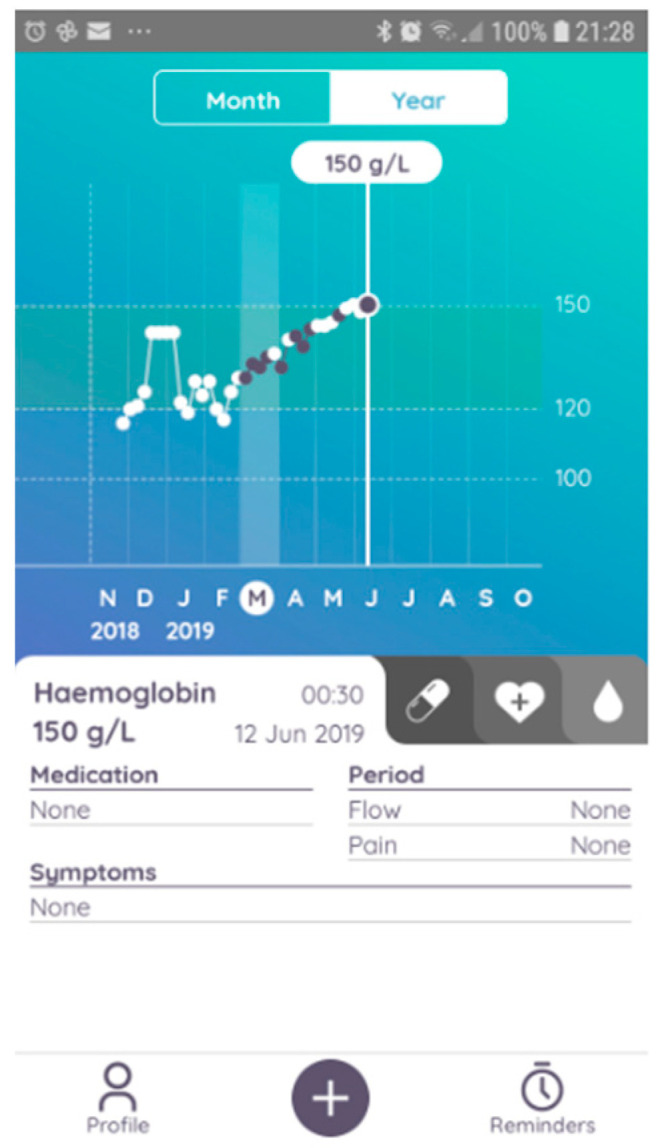
A screenshot taken from Luma’s associated mobile application. Trends in results are displayed, and annotations can be added. Image used with written permission from Entia Ltd., London, UK.

**Figure 3 diagnostics-14-00232-f003:**
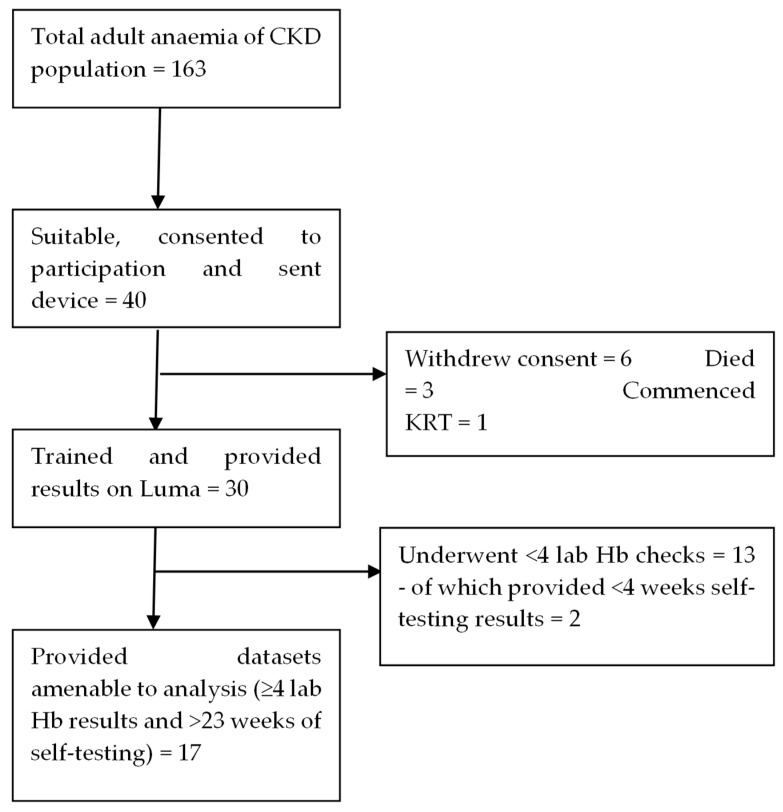
Consort diagram for study participants. Hb: haemoglobin, CKD: chronic kidney disease.

**Figure 4 diagnostics-14-00232-f004:**
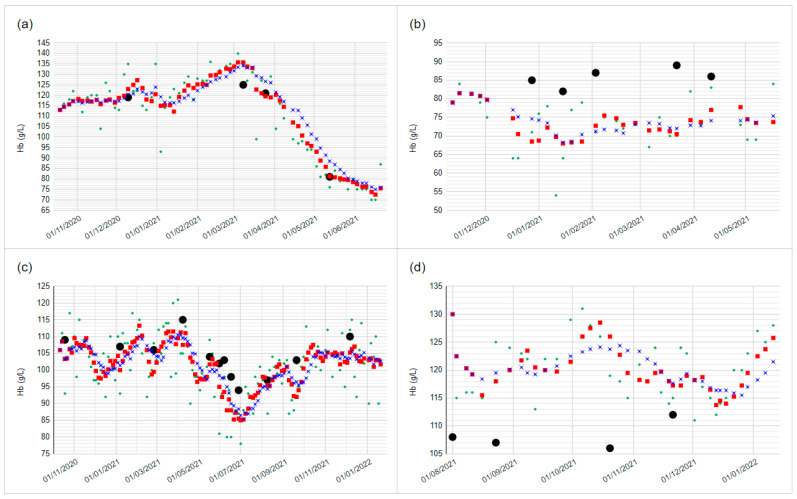
Examples of participants with various patterns of variability in their Luma testing. Vertical axis gridlines are at every month. (**a**) Low Luma testing variability. (**b**) High Luma testing variability. Note the large spread of Luma results throughout. (**c**) Average Luma testing variability. (**d**) High variability improves to low variability. Note the improving correlation with lab values. **Green dots:** Raw Luma Hb results, **Black dots:** Lab Hb results, **Red squares:** 4 point moving average (4PMA) of Luma Hb results. **Blue crosses:** 8PMA Luma Hb results. Hb: haemoglobin.

**Table 1 diagnostics-14-00232-t001:** Summary of Luma use and performance characteristics. Hb: haemoglobin, 8PMA: 8-point moving average, CV: coefficient of variance.

Luma Study Population	N = 17
Patients testing for >6 months	16 (94.1%)
** Of which: **	
Patients testing for >12 months	2 (11.8%)
Patients testing ≥ 94% of recommended twice-weekly frequency	6 (35.3%)
Patients testing > 60% of recommended twice-weekly frequency	14 (82.4%)
Mean tests/wk	1.6
Mean difference Luma Hb vs. Lab Hb: Raw 8PMA	−1.95 g/L (95% CI −18.93 to 15.02) 0.40 g/L (95% CI −0.4 to 1.2)
% Luma results > 10% different from Lab Hb: Raw 8PMA	30.97% 17.70%
Lab Hb vs. Luma Hb: CV	8.31%
Bias	−2.04%

**Table 2 diagnostics-14-00232-t002:** Summary of patient end of trial questionnaire.

Question	Mean Score 1 = Very Poor, 5 = Very Good
How would you rate your in-person training?	4.7/5
How would you rate your online training?	4.9/5
How would you rate the training materials provided for Luma?	4.7/5
How would you rate the service provided for Luma?	4.8/5
How would you rate your experience of the Luma Haemoglobin tracker app?	4.7/5
How would you rate your experience of performing tests with Luma?	4.6/5
How would you rate your overall experience of Luma?	4.9/5
Did you use the Luma Haemoglobin tracker app while performing tests with Luma?	Yes: 66%
Would you recommend Luma to someone with the same health condition?	Yes: 100%
How does Luma compare to the previous treatment pathway or your health condition?	Prefer Luma pathway: 100%

## Data Availability

All the data for this study are available publicly at https://doi.org/10.5061/dryad.0vt4b8h46 (accessed on 26 January 2023).

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
