# Peer review of "Home Haemoglobin Monitoring for the Titration of Erythropoietin-Stimulating Agents in Chronic Kidney Disease: A Pragmatic Pilot Trial"

_diagnostics, 2024, doi:10.3390/diagnostics14020232_

Round 1

Reviewer 1 Report

Comments and Suggestions for Authors

The manuscript describes the results of a pilot trial that used a point-of-care testing device and an associated eHealth pathway for home monitoring of erythropoietin stimulating agents therapy in patients with CKD. The trial ran from August 2020 to March 2022 in a single UK renal center. Participants were mailed a POCT device and trained remotely to self-test their Hb levels twice weekly for up to 1 year. The results were shared with healthcare professionals via a mobile application. The percentage of POCT device results differing by >10% from lab results was 30.9%, dropping to 17.7% using an 8-point moving average. The study found that the home self-testing with the POCT device yielded comparable results to traditional ESA monitoring conducted by HCPs. The research design appropriate and the conclusions supported by the results. I suggest that the author add content on mobile application so that readers can have a clearer understanding of the design of the POCT device. This manuscript deserves publication.

Reviewer 2 Report

Comments and Suggestions for Authors

This study has the attractive title and scientific richness. In other words, the present manuscript could be publishing in “diagnostic” after minor revision.

1-      First of all, old references (before 2018) must be removed in the main text and updated references substituted.

2-      The abstract is good written but the novelty of the study was missed, please expand it.

3-      Future prospective of study must be highlighted in the main text?

4-      In conclusion section some additional data (277-282 lines) must be removed to result part.

5-      There is no need for presentation of “Staff feedback on the Luma pathway” in the Table.3. Please insert the table.3 content in the main text.

6-      The advantages and drawback of the used method is missed in the main text.

Comments on the Quality of English Language

Minor editing of English language required

Reviewer 3 Report

Comments and Suggestions for Authors

The paper describes the results of a year of use of a POCT device with its associated eHealth pathway in the home monitoring of ESA therapy. It used a CE-marked POCT device designed for self-testing, which could 72 measure Hb from a drop of capillary blood obtained via a finger prick.

The paper is well organized, and the length is appropriate. The title is chosen correctly, but the abstract doesn’t provide sufficient information to give a clear idea of the paper's aim.

The study methods are appropriate, and the data are valid. The results are well highlighted, and the conclusions are adequate.

The references are relevant and correctly chosen, and related work is discussed and cited appropriately; however, I think other works should be considered.

The graphical representations in Fig 3 are difficult to visualize (perhaps due to the choice of colors).

Different time intervals are also chosen for the five performances. The data obtained at the same interval should be presented for a clearer picture.

Reviewer 4 Report

Comments and Suggestions for Authors

Results of a pilot trial on the applicability of a poin-of-care-testing device measuring hemoglobin values in a drop of capillary blood is presented in this publication. The target group of the study is patients with chronic kidney disease. Measurements made with a CE-marked POST device were compared with those obtained by specialized laboratory measurement equipment. The study was conducted over a period of approximately 20 months and the generalizations are based on the results obtained from seventeen participants returning comparable data sets, including ≥ 4 laboratory Hb measurements and self-measurements (as prescribed twice weekly) for >5 months. The abstract and analysis in the Introduction section are well written and consistent with the presented research and results in the article. Reference sources are relevant to the content and cited at appropriate places in the text.

Some comments and remarks:

1.     Applying an 8-point-moving-average procedure reduces the degree of deviation of real measurements from laboratory ones, but this does not change the fact that personal self-measurements differ significantly from reference values. The time delay between a reference measurement and a series of averaged individual measurements should be specified.

2.     Irregularity is observed when reporting and comparing data from laboratory tests and Luma tests, e.g. fig. 3a (break in lab tests for more than 3 months). It is not clear what the correlation is with the potential referent Hb values during this period.

3.     The authors claim that the comparable dataset included ≥4 lab Hb measurements, but in fig. 3d presents data with only 2 laboratory measurements.

4.     Claims related to the advantages of the proposed approach using a POCT device for Hb control do not correspond to the steadily decreasing number of participants, also of patient tests during the measurements.

Round 2

Reviewer 4 Report

Comments and Suggestions for Authors

I am satisfied with the authors' responses. I have no further questions or comments.